# Spectral Shift and Split of Harmonic Lines in Propagation Affected High Harmonic Generation in a Long-Interaction Gas Tube

Jozsef Seres [1],[*], Enikoe Seres [1], Carles Serrat [2], Thanh-Hung Dinh [3], Noboru Hasegawa [3], Masahiko Ishino [3], Masaharu Nishikino [3] and Shinichi Namba [4]

1   Institute of Atomic and Subatomic Physics E-141, Vienna University of Technology, Stadionallee 2, 1020 Vienna, Austria; enikoe.judit.seres@tuwien.ac.at
2   Department of Physics, Polytechnic University of Catalonia, Colom 11, 08222 Terrassa, Spain; carles.serrat-jurado@upc.edu
3   Kansai Institute for Photon Science, National Institutes for Quantum Science and Technology (QST), Kizugawa 619-0215, Japan; dinh.thanhhung@qst.go.jp (T.-H.D.); hasegawa.noboru@qst.go.jp (N.H.); ishino.masahiko@qst.go.jp (M.I.); nishikino.masaharu@qst.go.jp (M.N.)
4   Graduate School of Advanced Science and Engineering, Hiroshima University, 1-4-1 Kagamiyama, Higashihiroshima 739-8527, Japan; namba@hiroshima-u.ac.jp
*   Correspondence: jozsef.seres@tuwien.ac.at

**Abstract:** While generating high harmonics in long media of helium gas, at certain laser intensities and chirp, the spectral shift and split of the harmonic lines were experimentally observed, sometimes exceeding one harmonic order. Beyond reporting these results, numerical simulations were performed to understand the phenomenon. A 3D propagation model was solved under the strong field approximation. According to the simulations, the distortion of the laser beam profile during propagation and the consequently accused change in the conditions of phase matching are responsible for the observations. The observed phenomena can be an excellent tool to produce tunable narrow band harmonic sources covering a broad range around 13.5 nm for spectroscopy and for seeding X-ray lasers, and to understand non-desired detuning of the seed wavelength.

**Keywords:** high harmonic generation; light–atom interaction; non-linear propagation; strong-field physics

## 1. Introduction

Using ultrashort and intense near-infrared (NIR) laser pulses is a relatively convenient way to produce coherent short wavelength ultrashort pulses by the generation of high-order harmonics (HH) in the interaction between the NIR laser pulse and rare gases. The generated spectra can be composed from a few harmonic lines to thousands. Such a broad spectral range and ultrashort pulse duration opened the gate for wide variety of applications, including time-resolved extreme ultraviolet (EUV) and X-ray spectroscopy, high-resolution microscopy and EUV lithography. To improve the efficiency of the HH generation process for better applicability, different phase matching schemes, different illumination geometries [1] such as semi-infinite gas cells [2–6], and X-ray parametric amplification (XPA) [7–9] were successfully utilized. To understand their usefulness and limits, several experiments were conducted to study the beam profile [5,10–12] and spectral properties, specifically the harmonic line shape and spectral shift [4,5,13–15] of the HH radiation. In both cases, complex behaviors were observed. Inspecting the beam profiles and their interference effects, the role and importance of the short and long electron trajectories and the quantum path interferences in the generation process have been resolved [10–12,16–19]. The spectral shift of the harmonic lines from their expected spectral position was also

recognized. Beyond being an interesting phenomenon, spectral shift has practical importance because this effect can be used to produce a tunable source of harmonics [20,21] for spectroscopy or seeding X-ray lasers [22,23]. For tuning the HH source wavelength to that of the X-ray laser, the tuning of the wavelength of the pump laser would be an obvious option [15,23]. However, this may not always be feasible, as is the case with glass lasers, which have a narrow gain bandwidth. Other tuning options include adjusting the initial chirp [15,24] of the laser pulse, as well as modifying factors such as gas pressure, gas jet position, or laser intensity [20]. In addition, spectral shift of the harmonic lines can have undesirable consequences, as it can result in the detuning of the harmonic line from the required seed wavelength of the X-ray laser. When applying phase-matched HH schemes, it is essential to consider that the actual wavelengths of the harmonic lines may not be integer divisors of the pump laser wavelength [1].

The shift of the harmonic lines also reveals the underlying physical processes of HH generation. Blue or red shifts can be caused by the presence of XPA [25], and a blue shift can also be produced by the blue shift of the generating laser beam during propagation and nonlinear interaction in the gas medium. Such spectral shifts can be a limiting factor in the coherent buildup of the harmonics since they can negatively affect the generation efficiency [14] and the implementation of various phase-matching schemes.

The current work aims to produce an intense and tunable HH source, especially at 13.5 nm (~59th order of the 800 nm drive laser), a spectral region where HH can contribute to efficient extreme ultraviolet (EUV) lithography and the seeding [23] of X-ray lasers. In the present study, the drive laser traveled in a 20 mm long gas tube. The nonlinear propagation of the driving laser pulse considering the change of its beam profile and spectral shape, together with the phase-matching conditions produce a wildly tunable and bright HH source at around 13.5 nm.

After describing the experimental conditions and the theoretical model used for simulations, we examine the spectral shift and split of the harmonic lines due to the initial chirp of the pump laser pulse and the position of the laser focus in the long gas medium. The experimental results are compared with simulations, attributing observed effects to nonlinear propagation, gas ionization, and the resulting blue shift and beam profile changes in the laser pulses.

## 2. Methods Employed in Experiments and Simulations

### 2.1. Experimental Setup

The experiment was carried out at QST Kansai. The NIR drive laser was a Ti:Sapphire laser (center wavelength: 800 nm, pulse duration: 40 fs, beam diameter: 20 mm, maximum pulse energy: 80 mJ, repetition rate 10 Hz). The pulse duration was measured by a SPIDER, and the measured compressed pulse is presented in Figure 1c.

The laser was focused onto the gas target by a lens with a focal length of 4 m. The beam diameter (FWHM) was ~220 μm, and the Rayleigh length was 40 mm (both measured) supporting the measurements, with peak intensities of ~2.2 PW/cm$^2$ for 52 mJ and ~3.4 PW/cm$^2$ for 80 mJ pulse energies. The gas target was a Mo tube that was 20 (40 and 80) mm in length with a diameter of 2.9 mm. A photo of the 80 mm long gas target is seen in Figure 1b. The gas was supplied by a fast solenoid valve with various stagnation pressures. The valve opening time of ~5 ms served to limit HH absorption due to the gas. The gas target is illustrated in Figure 1a with the focused beam and the formed plasma. The HH spectra were measured with a grazing incident spectrometer with a spherical mirror, enabling us to estimate the beam divergence. Fundamental laser light was blocked by a thin Zr filter.

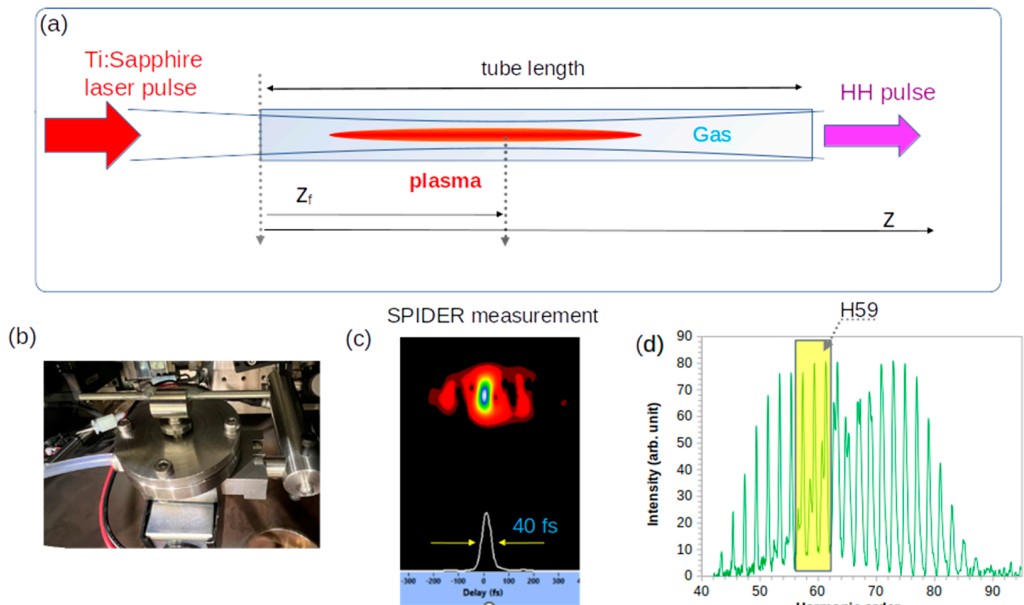

**Figure 1.** (**a**) The laser beam was loosely focused into the long tube. Plasma was formed at around the focus to generate harmonics. The position of the focus $z_f$ is measured from the entrance of the tube. (**b**) Photo of the 80 mm long tube connected to the fast solenoid valve. (**c**) SPIDER measurement of the laser pulse and (**d**) a typical measured spectrum. The highlighted harmonic lines at around 13.5 nm (59th harmonic) are examined in detail.

### 2.2. Theoretical Model

In order to describe the nonlinear propagation of the laser pulse in the long gas tube and the generation and propagation of HH pulses, a system of coupled differential equations for the electric fields of the laser and the harmonics was considered as in [26]:

$$\partial_\xi E_l(\xi, \tau) - \frac{c}{2} \nabla^2_\perp \int_{-\infty}^{\tau} E_l(\xi, \tau') d\tau' = \frac{-1}{2c} \int_{-\infty}^{\tau} \omega_p^2(\xi, \tau') E_l(\xi, \tau') d\tau' \\ \frac{-\zeta^{(1)}}{c} \partial_\tau [1 - n_e(\xi, \tau)] E_l(\xi, \tau) - \frac{W_b}{2\epsilon_0 c} \frac{\partial_\tau n_e(\xi, \tau)}{E_l(\xi, \tau)} \tag{1}$$

$$\partial_\xi E_h(\xi, \tau) - \frac{c}{2} \nabla^2_\perp \int_{-\infty}^{\tau} E_h(\xi, \tau') d\tau' = -\alpha_h E_h(\xi, \tau) - \frac{1}{2\epsilon_0 c} \partial_\tau P_h[E_l(\xi, \tau)] + c.c. \tag{2}$$

The second terms on the left-hand side of Equations (1) and (2) account for beam divergence considering cylindrical symmetry. On the right-hand side of Equation (1), the terms describe the free-electron and atomic dispersion, and the effect of the ionization. The right-hand side of Equation (2) includes the reabsorption of the generated harmonics in the gas during propagation and the generation term. $E_l$, $E_h$, $\omega_p$, $W_b$, $\epsilon_0$, c, $\alpha_h$, $P_h$, and *c.c.* represent the electric field of the laser and harmonics, the plasma frequency, the ionization potential, the speed of light, the absorption coefficient of the harmonics, the polarization, and the complex conjugate. A coordinate system moving with the speed of the light is introduced as $\tau = t - z/c$, $\xi = z$, and the equations are solved numerically, as detailed in [27]. The Keldysh parameter considering the parameter range of our study is well below the unity (~0.3–0.4), and therefore, we are in the condition of efficient tunneling [28]. In order to compute the ionization rate in the tunnel regime, we resort to the Ammosov–Delone–Krainov theory [29]. If a broader parameter range were to be considered, different ionization regimes could be readily incorporated into the propagation calculations [30,31].

## 3. Spectral Shift and the Spectral Shape of the Harmonic Lines
### 3.1. Dependence on the Initial Chirp of the Laser Pulse

It is expected that the laser pulses interact nonlinearly with the long gas medium during propagation, which alters the pulse shape, the spectral shape, and the beam profile

of the laser pulses. These changes affect the generated harmonic spectra. In one of the experimental series, the initial chirp (group delay dispersion: GDD) of the laser pulse was varied, while the laser beam was focused near the middle of the 20 mm long gas tube ($z_f$ = 7 mm). The used pulse energy of the laser pulses was 52 mJ. In the experiments, the chirp was changed by changing the distance between the gratings of the pulse compressor. The results at harmonic order around H59 are plotted in Figure 2a and for few GDDs, the spectra and beam profiles can be seen in Figure 2c,d. At GDD < 0, a small blue shift proportional to the GDD can be observed. It is followed by a well-defined and strong spectral blue shift, and a change in the shape of the harmonic lines starts at around GDD = 0 fs$^2$ and remains until about +1000 fs$^2$. After that, narrower and cleaner harmonic lines are generated, and their spectral positions regress back to the non-shifted position. The beam profiles of Figure 2d exhibit some interesting behaviors. In both cases, a large divergent and weak component can be seen together with a much stronger low divergent beam. These large divergent components appear like thick (blue) vertical lines above and below the small and strong beams. They are very weak and hardly observable. Such a beam structure was also observed earlier [9,12] and explained by the contribution of different electron trajectories (long and short) in the generation process or the contribution from different radial parts of the generating laser pulse [11]. Beyond the difference in the divergence, it can be observed that the small divergent intense beams are somewhat blue-shifted at +600 fs$^2$ and somewhat red-shifted at −1400 fs$^2$ relative to the large divergent component.

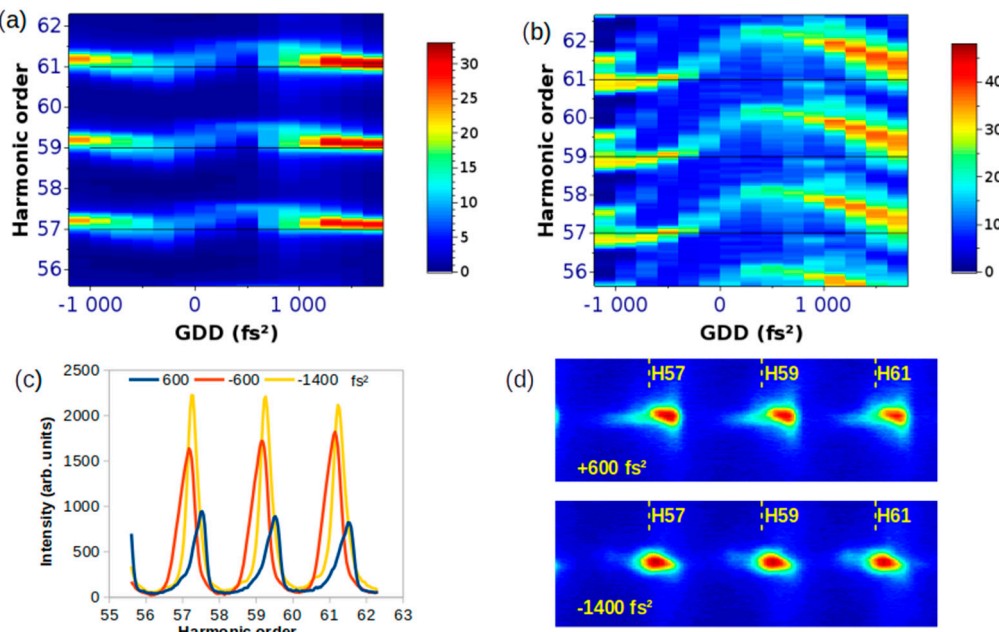

**Figure 2.** Both the line shift and line shape of the (**a**) measured and the (**b**) simulated spectral lines are strongly affected by the chirp of the pump pulse. The color bars are in arbitrary units. (**c**) Measured harmonic lines and (**d**) spectrally resolved beam profiles at around H59 for different initial chirps of the driver laser pulses. Dashed lines of (**d**) indicate the non-shifted positions of the harmonic lines.

To gain deeper insights into the behavior of the generated harmonics, simulations were conducted by numerically solving Equations (1) and (2). This approach proves invaluable as it allows for the calculation of laser pulse propagation within the extended gas tube, enabling the tracking of spatial variations in both the longitudinal and radial directions of the laser pulse and the associated harmonic generation. Such detailed analysis is challenging to achieve experimentally. The generated harmonics at H59 are plotted in Figure 2b as a function of the initial chirp of the laser pulse. The harmonics are from the end of the tube and radially integrated to be comparable to the experiment in Figure 2a. The

stagnation gas pressure for the gas valve was 0.3 bar in the experiments, which corresponds to about a 20 mbar pressure in the interaction range based on a comparison between the measurements and the simulations. In agreement with the observed effects in the experiment, the simulations accurately reproduce parallel behaviors, including spectral shifts and intensity changes in the harmonic lines. The chirp of the pump pulse strongly affects the shift and the split of the harmonic lines. Between values of the GDD of 0 fs$^2$ and +1000 fs$^2$, a strong blue shift and spectral shape distortion can be observed. The spectral shifts of the simulations are somewhat larger than in the experiments. The reason for this can be that the peak laser intensity of the laser pulse in the experiments was somewhat smaller than in the simulations, because the pulse shapes (temporal and spatial) were somewhat different.

To understand what is actually happening in the gas tube, the spatial evolution of the harmonics spectrum and the spectral and radial change of the laser pulse are calculated and presented in Figure 3a–c at four values of GDD (+200 fs$^2$, +600 fs$^2$, +1000 fs$^2$ and +1400 fs$^2$) within the range where the harmonic spectra experienced large blue shift and shape distortion. These effects are the consequence of the strong ionization of the gas medium by the laser pulse and the feedback of the ionization to the pulse propagation, which was considered by the last term of Equation (1). The first observation is that at an almost fully compressed laser pulse, +200 fs$^2$, the laser spectrum becomes strongly blue-shifted (~4%), splits at the first 5 mm part of the tube, and remains so later. Within the first 5 mm propagation, the beam profile also develops into a flat-top-like shape with some ring structures. Both are the consequence of the large nonlinear effect of the ionization of the gas medium [32,33].

At increased chirp, both the blue shift of the laser pulse and the distortion of the beam profile are getting smaller and almost disappear at +1400 fs$^2$ because of the much lower ionization rate at the lower laser intensities of the chirped pulses. It could be expected that the generated harmonic spectra follow the mentioned blue shift and split of the laser pulse spectrum; however, it only happens partly. The largest blue shift of ~2.4 harmonic order can be expected from the ~0.04-harmonic-order blue shift of the laser pulse at +200 fs$^2$, however the blue shift of H59 is ~1.1 harmonic order (at the position of H60.1) and no split can be observed. Contrarily, at +600 fs$^2$, H59 splits while the laser spectrum does not split. The blue shift of the stronger part is about the same (~1.1) as for +200 fs$^2$ chirp, which is close to the expected ~1.2 harmonic order calculated by the ~0.02 harmonic order blue shift of the laser spectrum. The harmonic lines split even at +1000 fs$^2$ chirp and for both +1000 fs$^2$ and +1400 fs$^2$ chirps, the blue shifts start to decrease following the blue shift of the laser pulse. The observed differences at small chirp values can be understood as the effects of the phase-matching conditions during propagation in the gas medium because at low chirp (short pulse), the beam profile distortion is large, which increases the Gouy-phase contribution, as studied in detail in [33], and temporal phase-matching can be achieved at a certain point of the generation medium. This phenomenon can be observed in the panels of Figure 3d. At +200 fs$^2$ chirp, the phase-matching increase in the harmonic yield can only be observed at the beginning (z < 2 mm) of the medium and at around z = 17 mm. At other parts of the medium, the increase in the yield is nearly linear, which is a clear consequence of the blue shift in the laser beam [14]. At +1000 fs$^2$, more temporal phase-matching positions can be recognized, and H59 becomes strong, while at +1400 fs$^2$, at the end of the medium, the effect of the saturation can be recognized.

### 3.2. Independence of the Focus Position in the Gas Tube

In the previous section, it was observed that the intensity, spectral shift, and shape of the generated harmonics were strongly dependent on the initial chirp of the laser pulse. This was experimentally demonstrated, and the simulations showed the major role of the propagation nonlinearity of the laser pulse, both temporally and spatially. Because of the strong nonlinear effects during propagation, the position of the focus ($z_f$) within the long medium can be important.

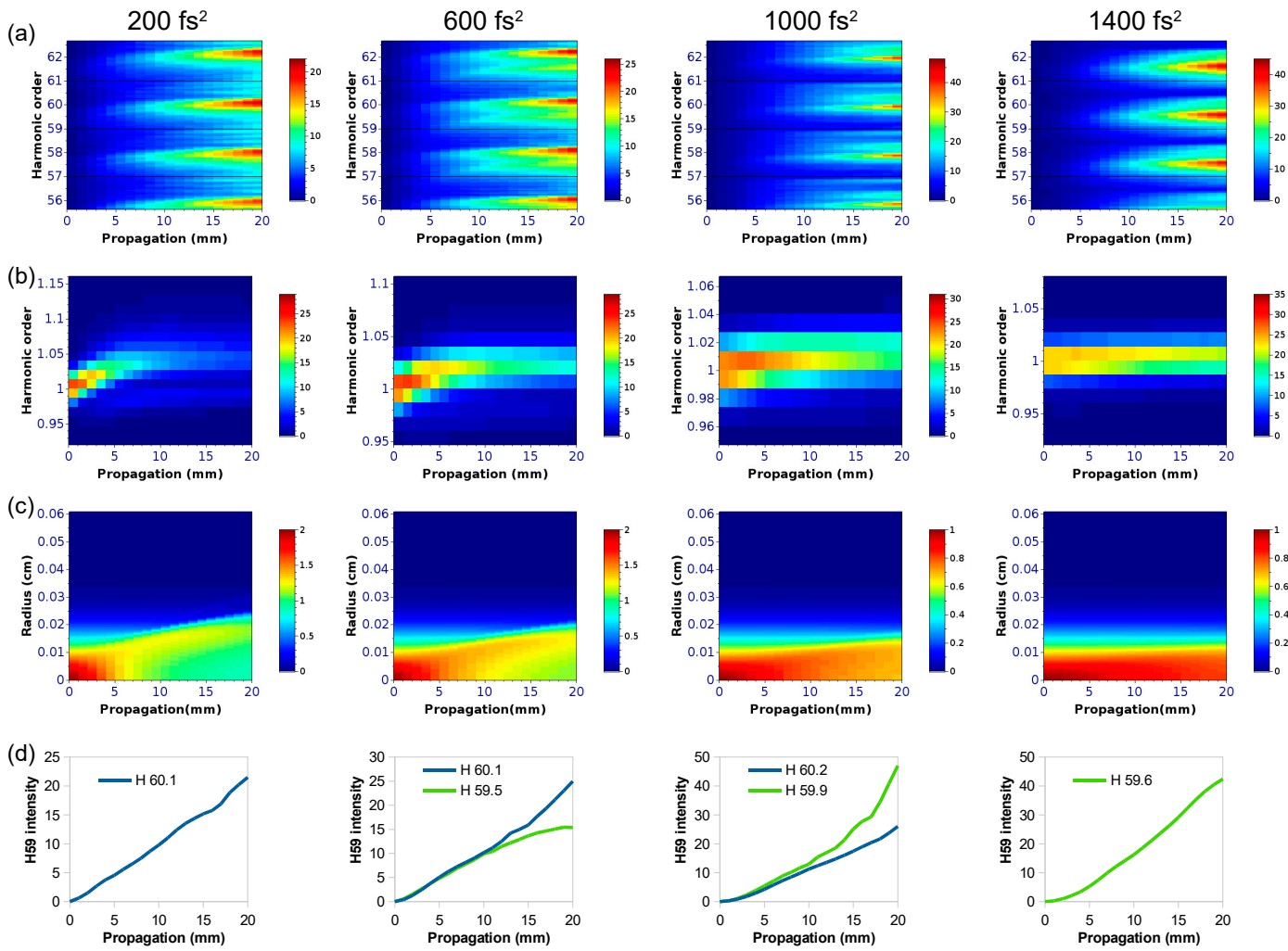

**Figure 3.** The simulations show the evolution of (**a**) the harmonic lines around H59, (**b**) the laser spectrum, and (**c**) the beam profile during propagation. (**d**) Evolution of the peak intensity of the blue shift and split of H59 at certain spectral position given in harmonic order.

Experiments were performed to address this question, and the results are presented in Figure 4. The focus position was scanned from the beginning ($z_f$ = 0 mm) of the 20 mm long gas tube. The definition of the $z_f$ is shown in Figure 1a. For 52 mJ pump, Figure 4a, the initial chirp was chosen to be −800 fs², where strong splits of the harmonic lines were observed. In the cases of the spectral shape and the beam profile, no dependence on the focus position was observed. The intensity of harmonic lines has only a weak dependence. Choosing a larger 80 mJ energy of the laser pulse, an energy for which much stronger nonlinear effects can be expected, a much stronger split of the harmonic lines occurs, although no essential dependence of the harmonic signal (intensity, spectral shape, and beam profile) on the focus position was observed, as presented in Figure 4b. It is possible only to recognize a small spectral shift. We can conclude that the choice of the focus position essentially does not affect the spectral position and shape of the harmonic lines.

### 3.3. The Spectral Shape of the Harmonic Lines Strongly Affected by the Pump Energy

As it was visible in Figure 4, both the intensity of the generated harmonics around H59 and also the spectral shift and shape of the harmonic lines were weakly affected by the position of the focus in the long gas tube. This observation is further supported by the results presented in Figure 5, where the focus was placed 21 mm before the entrance of the gas tube. Other experimental and simulation parameters were the same. One can compare the simulation in Figure 2b to the simulation in Figure 5a at a lower pump energy

(52 mJ). The simulations basically show the very similar behaviors of GDD dependence. The simulation at the larger pump energy of 80 mJ predicts a somewhat larger spectral shift of the harmonic lines compared to 52 mJ pumping. In both cases, the harmonic lines are split at around GDD = 0 fs$^2$, while they are more intense and continuously tunable at positive GDD.

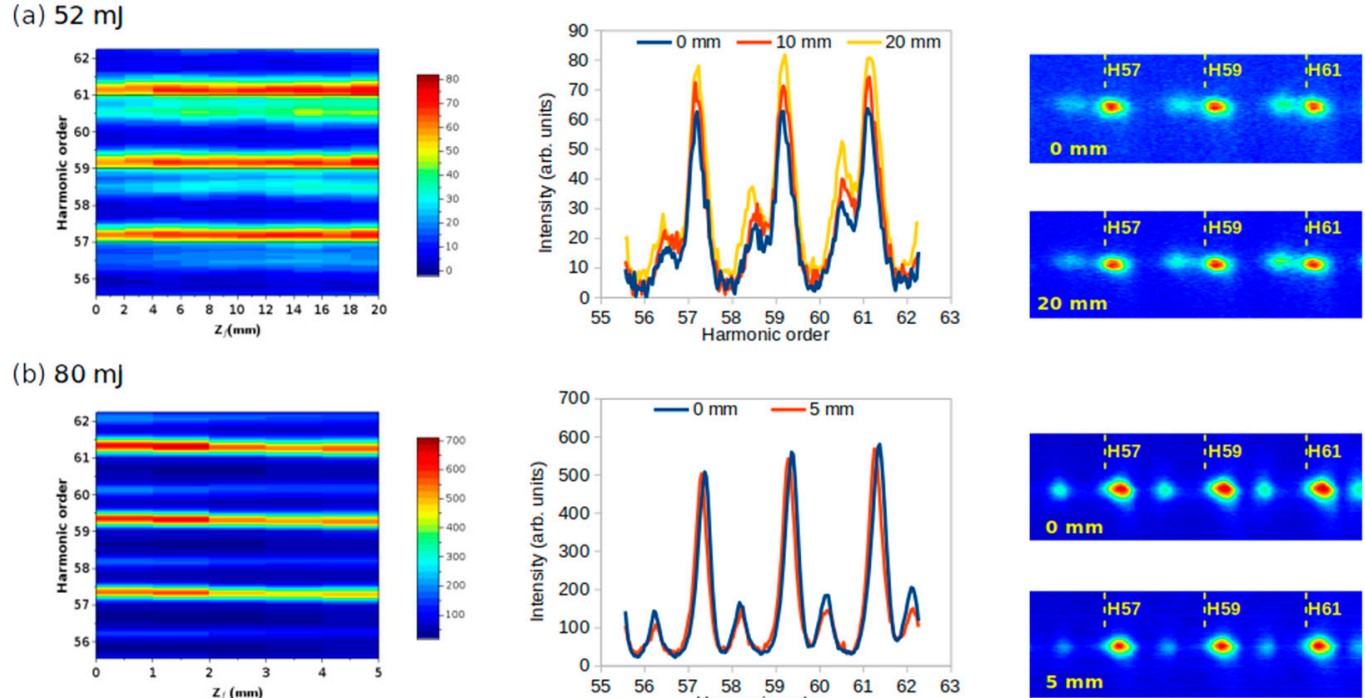

**Figure 4.** The spectral positions and shape/split and beam profiles of the harmonic lines affected only slightly by the focus position in the gas tube both at (**a**) 52 mJ and at (**b**) 80 mJ pump energies. Dashed lines of the right column indicate the non-shifted positions of the harmonic lines.

We are here mainly interested in a tunable and intense harmonic source for spectroscopy and seeding X-ray lasers. Thus, the suitable range of positive GDD was examined experimentally in Figure 5b at both 52 mJ and 80 mJ pump energies. While the simulations predicted larger values, about one harmonic order broad tuning range, the experiments show about a 0.5 harmonic order tuning possibility, and this essentially does not change at higher pump energies. The split of the harmonics line, however, is much larger at 80 mJ pump compared to 52 mJ. Strong blue-shifted harmonic lines can be observed together with weaker red-shifted lines, and both regress back to their original harmonic position at an increased chirp of the pump laser pulse. The split of the harmonic lines can be so large that the red-shifted harmonics line almost merge with the blue-shifted previous harmonics lines, as can be seen on the spectrally resolved beam profiles in Figure 5c.

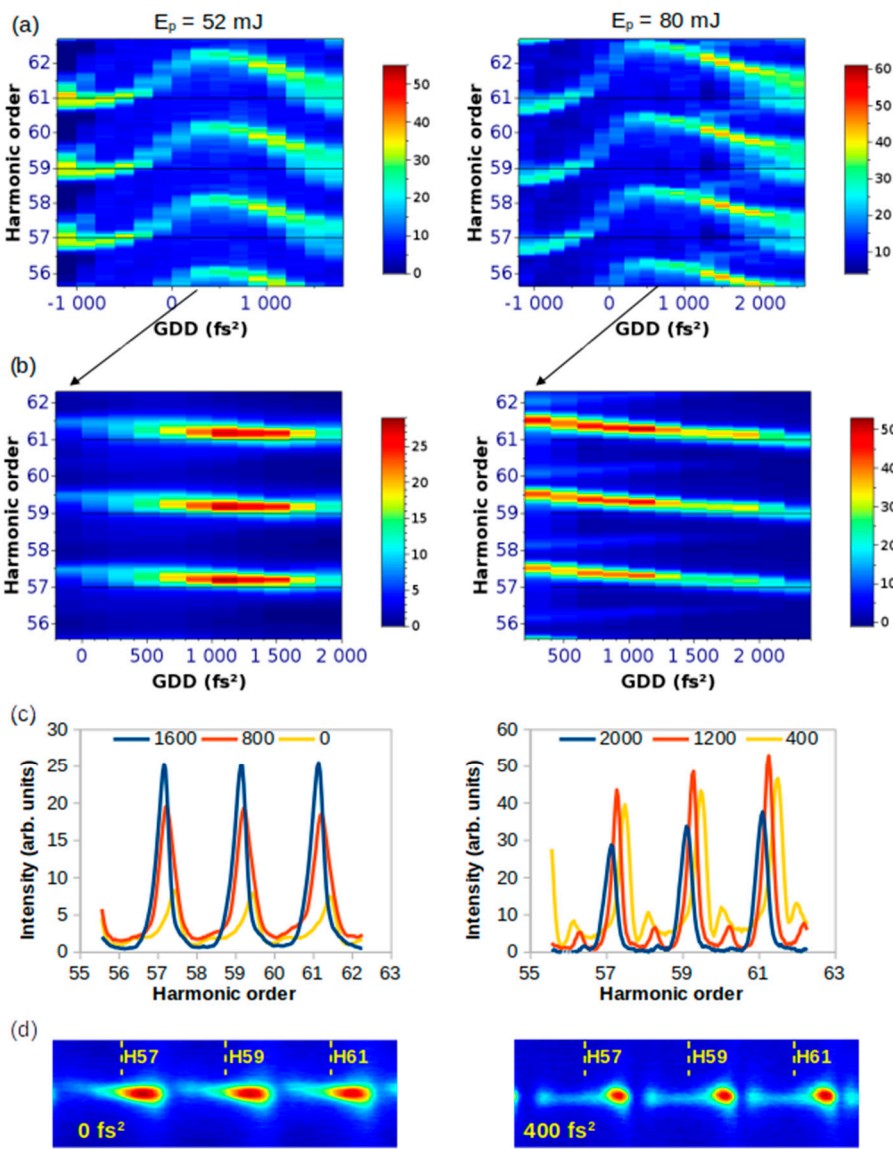

**Figure 5.** (**a**) Simulations of HH generation for the chirps of the driving laser pulses being scanned in a broad interval at two pump energies, 52 mJ and 80 mJ, as indicated. (**b**) Measured harmonics at around H59 in the positive GDD range, where the harmonics are intense and well tunable. (**c**) Spectra and (**d**) beam profiles from (**b**) at few initial chirps (fs$^2$) indicated in the labels. Dashed lines of (**d**) indicate the non-shifted positions of the harmonic lines.

## 4. Conclusions

In the present study, high harmonic generation in a 20 mm long gas medium was examined both experimentally and theoretically. At certain experimental conditions, a strong blue shift of the harmonic lines was observed accompanied sometimes by a strong spectral distortion and even a spectral split of the harmonic lines exceeding one harmonic order. To understand these observations, we employed a theoretical model—a 3D propagation model—that describes the propagation and high harmonic generation in the gas medium, considering both longitudinal and radial directions. This model was numerically solved under the strong field approximation. The simulations revealed that, the blue shift and split of the harmonic lines were partially caused by the blue shift and spectral split of the laser beam during propagation in the gas, which was observed under conditions when the driving laser pulses were positively chirped in the range between 0 fs$^2$ and 1000 fs$^2$. The distortion of the laser beam profile during propagation leading to changes in the conditions of phase-matching played an essential role and primarily caused the observed phenomena.

The nonlinear propagation of the driving laser pulse considering the change of its beam profile and spectral shape, together with the phase matching conditions, produced a bright HH source, which was widely tunable at around 13.5 nm (~59th order of the 800 nm drive laser). The beam divergence of the generated harmonic beam remained small, within the $0.11 \pm 0.02$ mrad range (defined by FWHM), even under divergent experimental conditions, as shown in Figure 2d, the right column of Figure 4, and in Figure 5c. The energy of the generated EUV pulse of the 59th harmonic in Figure 5c was approximately 47 pJ. A seed beam with a suitable small divergence and large pulse energy is essential for seeding X-ray lasers [34]. The observed phenomena offer a promising tool to produce tunable narrow-band harmonic sources, especially at 13.5 nm, a spectral region where HH can contribute to efficient extreme ultraviolet (EUV) lithography and seeding of X-ray lasers. For practical applications of a 13.9 nm Ni-like Ag plasma X-ray laser, the 57th order harmonic is most suitable when considering the use of a Ti:sapphire laser with a fundamental wavelength of 792.3 nm.

**Author Contributions:** Conceptualization, S.N. and T.-H.D.; methodology, S.N. and C.S.; software, C.S.; formal analysis, J.S., E.S. and C.S.; investigation, T.-H.D., N.H., M.I., M.N. and S.N.; resources, S.N.; data curation, J.S.; writing—original draft preparation, J.S., E.S., C.S. and S.N.; writing—review and editing, J.S., E.S., C.S., T.-H.D., N.H., M.I., M.N. and S.N.; supervision, S.N.; funding acquisition, S.N. All authors have read and agreed to the published version of the manuscript.

**Funding:** This research was funded by Japan Society for the Promotion of Science, KAKENHI, JP20H00141, JP19K15402, and JP21H03750; JKA and its promotion funds from KEIRIN RACE; and the transmittance calibration of Zr thin films was performed at BL11D of the Photon Factory, KEK Japan (Proposal Nos. 2019G547 and 2021G525). This study was also funded by the Horizon 2020 Framework Programme (856415); European Metrology Program for Innovation and Research (20FUN01 TSCAC); Österreichische Nationalstiftung für Forschung (AQUnet).

**Data Availability Statement:** The data that support the findings of this study are available from the corresponding authors upon reasonable request.

**Acknowledgments:** Computation time was provided by the Supercomputer System, Institute for Chemical Research, Kyoto University. The authors are grateful to S. Kojima, M. Mori, K. Kondo, Y. Yamamoto, and F. Ito from KPSI-QST for their unparalleled technical support.

**Conflicts of Interest:** The authors declare no conflict of interest.

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
