# Peer review of "Spectral Shift and Split of Harmonic Lines in Propagation Affected High Harmonic Generation in a Long-Interaction Gas Tube"

_atoms, doi:10.3390/atoms11120150_

Round 1

Reviewer 1 Report

Comments and Suggestions for Authors

Seres et al  show the tunability performance of a HH beam at around 13.5 nm when the spectral phase of the driver laser is varied.  There is a strong interest in this wavelength range due to applications in EUV lithography , and so this work is worth considering for publication. The results are interesting, although not very novel in my view (see for example https://opg.optica.org/josab/fulltext.cfm?uri=josab-24-5-1138&id=132510). The experiments were accompanied by a 3D propagation model (2D +1 in fact, assuming radial symmetry), and includes some ionization and propagation effects pertinent to the calculation of the resulting HH beam.

In my view there are some aspects of the paper that need to be improved before is accepted for publication. Among them:

- The work in reference 20, which is quite relevant to the current work, also required tunability of the HH. This was achieved by tuning the center wavelength of the laser oscillator employing an etalon. This should be mentioned. Additionally, is missing a reference to the previous work from the same team, where they employed the same technique (https://link.aps.org/doi/10.1103/PhysRevLett.97.123901).

- Using the chirp to tune the HH also modulates the achievable EUV intensity, what is the specific advantage of changing chirp vs changing simply the center wavelength of the laser? Most of these systems are based on Ti:Sapphire so there should be some flexibility to do so, and in that case there would not be a correlation between tuned wavelength and power.

- The model includes ionization effects. In this regard it would be beneficial to talk about different ionization regimes depending on Keldysh parameter. In fact this was studied for HHG in https://link.aps.org/doi/10.1103/PhysRevLett.111.073901

The ionization model can be also used for studying the propagation of ultrafast pulses in long gas capillaries, as demonstrated in https://opg.optica.org/oe/fulltext.cfm?uri=oe-20-8-9099&id=231733

It would be interesting to mention this approach to the ionization effects modeling in the manuscript.

- The authors used a SPIDER to characterize the Fourier limited pulse when the compressor is set to 0 fs^2. It would have been useful to have information about the higher order dispersion terms of the compressor when is mismatched. 

- The theoretical model does not really reproduce the physics observed in the experiment. In particular, why Figure 5b does not show the data for negative GDD?

Reviewer 2 Report

Comments and Suggestions for Authors

The authors aim to shift harmonic line around H59 of IR beam corresponding to wavelength 13.5 nm near X-ray laser wavelength at 13.9 nm and for lithography at 13 nm. In this publication, the authors demonstrate that varying IR pulse chirp will shift harmonic lines around H59.

The authors present experimental studies of spectral shift and split of harmonic lines of high harmonic generated by a fs IR laser beam in a long gas tube filled with Helium. The experimental results are well reproduced by simulations taking into account propagation of IR laser beam in a nonlinear medium and ionization of Helium atoms due to high intensity (> 1 PW/cm2). The high harmonic beam propagation is also described with a source term corresponding to linear and nonlinear polarization. The atomic dipole is described by Strong Field Approximation (SFA) model. The phase-matching aspect is contained in the two coupled propagation equations.

The authors varied the laser IR pulse chirp and showed experimentally and with simulations how harmonic lines are detuned and split. Mainly, the harmonic lines follow the IR blue shift due to electrons They showed experimentally and with simulations that detuning and splitting do not depend on IR laser focus position in the gas medium. As expected, the detuning and the splitting increase with energy.

General comments:

The authors showed nice agreement between experimental results and simulations. IR laser chirp is generally varied to shift the harmonics but the value of detuning is not experimentally reproducible. This study is then very important to understand and predict the experimental harmonic shift when varying experimental parameters as chirp but also laser energy.  

Nevertheless, it will be interesting to understand the physical explanation of each experimental result, by splitting what is the result of IR pulse chirp, electrons emerging from ionization and self-phase modulation due to Kerr effect. It would have been interesting to measure the IR pulse duration for each chirp (or GDD) value by SPIDER: intensity estimation will be important to understand the lack of signal at 0 GDD due to ionization effect on IR beam propagation and on phase-matching detuning. We can clearly see on Fig. 3c the defocusing effect due to electrons on IR beam propagation profiles for GDD < 1000 fs2.

The intensity estimation is not consistent with Ref 24 publication: in this publication, the pulse duration is 80 fs (twice the pulse duration indicated in this paper) and the energy is 30 mJ (instead of 52 mJ in this paper) for nearly the same focus size and the authors give the same peak intensity (1PW/cm2). Could they give better estimation of the peak intensity? Is it possible to know ionization percentage at this intensity for Helium atoms? It will then be interesting to know at each part at the front edge of the pulse and the harmonic spectral shift will depend on the IR instantaneous frequency (w < w0 for negative GGD and w > w0 for positive GDD). Is the blue shift due to ionization preponderant? The authors should have discussed this point.

The study with different position of the focus inside the gas medium is performed at GDD = -800 fs2 where no simulations are shown in Fig. 3. Why the authors have chosen to show experimental data at this GDD value? As the Rayleigh range (40mm) is larger than the gas medium length (20mm), one can assume that the impact of focus position will be weak.

Finally, it lacks a study with different gas pressure in order to show if phase-matching effect is predominant to understand the detuning and the splitting of harmonics.

In conclusion the authors claimed that then can afford the right wavelength tuning to reach the X-ray laser line at 13.9 nm or for lithography at 13nm. It seems that it will be better to consider H57 detuning to reach a wavelength of 13.9 nm because the harmonic lines are mainly blue-shifted. For lithography, as claimed by the authors, a wavelength of 13 nm can be reached by shifting H59. Nevertheless, the authors do not give any estimation of number of photons for these harmonics or conversion efficiency as in Ref Lambert, G., Gautier, J., Hauri, C. P., Zeitoun, P., Valentin, C., Marchenko, T., ... & Sebban, S. (2009). An optimized kHz two-colour high harmonic source for seeding free-electron lasers and plasma-based soft x-ray lasers. New journal of physics, 11(8), 083033. It will be important to know if the energy for H57 or H59 is above Amplified Spontaneous Emission (ASE) to be injected into X-ray laser plasma for 13.9 nm emission.

Specific comments or corrections:

62: a publication plan will be appreciated

71: is it an estimation of focus size or a measurement?

74: it will be important to know the laser repetition rate (10 Hz as in Ref. 24?) and compare to the valve opening time. The sentence is not well written: the HH absorption by the gas is not suppressed (line 94) but limited

78: is the beam divergence important in this paper? Did the authors study the harmonic divergence versus chirp, IR beam focus position or IR laser energy? It could be interesting to discuss this part if one wants to use this harmonic beam for X-ray laser seeding.

117: how is defined the non-shifted position experimentally? Calibration of the spectrometer?

118: one cannot see clearly where is the weak component: is it the largest divergence part?

120: harmonic beam structure due to short and long trajectories are well explained in following reference: Catoire, F., Ferré, A., Hort, O., Dubrouil, A., Quintard, L., Descamps, D., ... & Constant, E. (2016). Complex structure of spatially resolved high-order-harmonic spectra. Physical Review A, 94(6), 063401.

Fig. 2: the author should give the same chirps and the same signal maximum for both Fig.2a and Fig. 2b. The author should add the divergence measurement for Fig. 2d as well as the non-shifted positions for H57, H69 and H61 (small vertical lines)

138: the authors should calculate the absorption length for 20 mb of Helium for H59 using cxro website, and compare it to the medium length and the blue-shift length.

Fig. 3a: why the harmonic lines are not calculated until the end of gas medium = 20mm?

182: “which is a clear consequence of the blue shift” instead of “which is a clear consequence the blue shift”

191: “the position displacement of the focus” instead of “the position of the focus”

Fig. 4: For an energy of 80 mJ, why the harmonic signal is shown for a gas medium of 5 mm length instead of 20 mm for an energy of 52 mJ? Could the authors add a vertical line for last column to show where is the non-perturbative position of each harmonic line?

Fig. 5: the authors should put the same scale for both energies in Fig. 5a. Why the GDD scan for E = 80 mJ is different than for E = 52 mJ in Fig. 5b?

237: For higher energy, the authors show that the splitting is higher and the energy in the split part is also higher. Have the authors estimated the loss percentage in those split parts?

282: “https://” instead of “hhttps://”

Regarding all these comments and questions, I recommend to reconsider this publication submission after major revision (control missing in some experiments).

Round 2

Reviewer 2 Report

Comments and Suggestions for Authors

As the authors follow the corrections and suggestions I had listed, I have no more comment.